# Propagation and control of congestion risk in scale-free networks based on information entropy

Huining Yan[1], Hua Li[2]*, Qiubai Sun[3], Yuxi Jiang[4]

**1** School of Electronic and Information Engineering, University of Science and Technology Liaoning, Anshan, Liaoning, China, **2** School of Business Administration, University of Science and Technology Liaoning, Anshan, Liaoning, China, **3** Asset Company, University of Science and Technology Liaoning, Anshan, Liaoning, China, **4** School of Economics and Management, Dalian Jiaotong University, Dalian, Liaoning, China

* lh1@ustl.edu.cn

**Data Availability Statement:** All relevant data are within the paper.

**Funding:** This research is funded by Natural Science Foundation of China (71771112) and Project of Liaoning Provincial Federation Social

## Abstract

To study the propagation pattern of congestion risk in the traffic network and enhance risk control capabilities, a model has been developed. This model takes into account the probabilities of five threats (the risk occurrence probability; the risk of loss; the unpredictability of risk; the uncontrollability of risk; the transferability of risk) in the traffic network to define the risk entropy and determine the risk capacity, analyze the mechanism of congestion risk propagation, and explore the impact of risk resistance, the average degree of risk capacity at intersections, and the degree of correlation on congestion risk propagation. Further, a control method model for risk propagation is proposed. Numerical simulation results demonstrate that the risk resistance parameter $\theta$ can inhibit the propagation of congestion risk during traffic congestion. The highest efficiency in controlling risk propagation is achieved when $\theta$ reaches a threshold value $\theta^*$. Furthermore, the average degree of intersection risk capacity $\alpha$ shows a positive correlation with $\theta^*$ and a negative correlation with control efficiency. However, the degree of association $\omega$ has a negative effect on risk propagation control, decreasing the degree of association between nodes aids in risk propagation control.

## Introduction

Recently, with the growth of the national economy, people are more willing to choose mobility vehicles for traveling, which increased pressure on intersections, making them more prone to congestion. Because intersections are interconnected, the network congestion risk propagates among them over time, resulting in an overall increase in congestion risk within the entire traffic network. This, in turn, negatively impacts orderly and stable social traffic. Therefore, studying the propagation behavior and control of congestion risk between individual intersections is of great importance for the smooth operation of urban traffic. Currently, most research on risk propagation is focused on corporate R&D networks [1, 2], public emergencies [3–8], power systems [9–12], supply chains [13–15], aviation operations [16–19], and similar areas. There are

Science Circles of China (L20BGL047). The funders had no role in study design, data collection and analysis, decision to publish, or preparation of the manuscript.

**Competing interests:** The authors have declared that no competing interests exist.

only a few studies on the propagation behavior and control of congestion risk in urban transport networks. Huang et al.examined the influence of travelers' behavioral characteristics on the propagation of road congestion risk using an improved infectious disease UAU–SIR model [20]. Hu et al.constructed SIR models to control risk nodes while considering urban congestion, effectively suppressing the scale of traffic accident risk contagion [21]. Chen et al.revealed how sudden incidents in metro operations can trigger the spread of crowding risks, leading to passenger strandings, line changes, panic, and stampede accidents [22]. Chen et al.also developed a weighted complex network to identify an optimal strategy for resisting congestion contagion risk [23]. Fei et al.analyzed congestion risk propagation characteristics by modeling the propagation speed of congested road sections [24]. Shan et al.proposed a greedy algorithm–based method for estimating the propagation path of traffic congestion and built a propagation network of weighted directed graphs to predict the propagation process of congestion between different network segments [25]. Cheng et al.identified the propagation paths of periodic congestion and analyzed their mechanisms using dynamic Bayesian networks, thereby alleviating traffic congestion at its source and blocking the propagation paths [26].

Most scholars have primarily used methods such as complex networks and contagion models to study the mechanisms of traffic congestion risk propagation. However, Priambodo et al.developed a method to predict the impact of road congestion on traffic conditions by analyzing spatial and historical data on traffic flows and employing statistical methods to establish relationships between traffic conditions (congestion or smooth flow) and traffic patterns [27]. Chen et al.attempted to model the congestion propagation phenomenon using a space–time congestion subgraph [28]. Predicting and controlling the spread of traffic congestion remains an ongoing challenge in most urban settings. Saberi et al.applied a transmission model of infectious disease spread in a population to a traffic congestion model and validated its effectiveness using large-scale data from six cities worldwide [29]. The congestion propagation modeling algorithm proposed by Nagy and Simon was the first algorithm to determine the propagation time expectation and the propagation probability of any propagation pattern using Markov chains [30].

In summary, existing research has focused on identifying and predicting congestion risk and using infectious disease models to explore the issue of congestion risk propagation mechanisms. However, relatively little research has been conducted to control the propagation of congestion risk. Moreover, classical infectious disease models only propagate the health or infection status of a node, ignoring that risk can accumulate at a node until it exceeds its tolerable range. Therefore, this study makes the following improvements to the risk contagion model based on the risk capacity proposed by Liu et al [1]: (1) adding indicators that quantify risk in terms of information entropy and (2) incorporating the degree of correlation between nodes into the model.

The study is organized as follows: in Chapter 2, a scale-free network is constructed to simulate the traffic network, using information entropy to quantify the risk of congestion at each intersection and calculate the risk capacity of each intersection. In Chapter 3, a model is developed to control congestion risk contagion by considering risk resistance and the degree of association. Chapter 4 analyzes the propagation mechanism of congestion risk through numerical simulations as well as the impact of the risk resistance parameter $\theta$, the average degree of intersection risk capacity $\alpha$, and the degree of association $\omega$ at intersections on the control of congestion risk. Finally, the full text is summarized in Chapter 5.

## Problem description and modeling

### Construction of the traffic network generation model

First, this study assumes that the traffic network is a complex network comprising numerous road intersections. In this network, the nodes represent intersections and the lines between

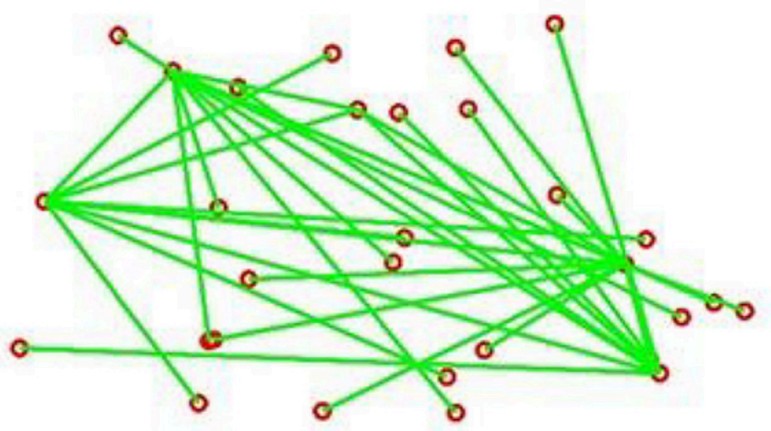

**Fig 1. Traffic scale-free network.**

two points represent edges, indicating the propagation path of congestion risk in the network. Considering the mutual role of road intersection risk propagation in reality, this study chooses the undirected graph G(V,E), where $V = \{1, 2, 3, \ldots, N\}$ is the set of all road intersections and $E = \{e_{ij} \mid i, j \in V\} \subseteq V \times V$ is the set of roads. $A = [a_{ij}]_{N \times N}$ is the adjacency matrix of the traffic network, where $a_{ij} = 1$ indicates that junctions $i$ and $j$ are connected and $a_{ij} = 0$ indicates that they are not yet connected. Second, people's choice of roads for travel is characterized by growth and meritocracy, with "growth" primarily reflecting the fact that travelers are constantly selecting intersections and thus adding them to the network. Starting with an initial network of $m_0$ intersections, a new intersection is introduced simultaneously and connected to $m$ existing intersections ($m \leq m_0$). "Meritocratic selectivity" indicates that travelers prefer certain roads for travel. Specifically, the closer an intersection is to its destination and centrally located, the more travelers it attracts. Here, the probability of a new intersection connecting an existing intersection $i$ is $\prod_i = (d_i + 1)/\sum_j(d_j + 1)$, where $d_i$ is the degree of the intersection $i$. Therefore, this study develops a propagation model for BA scale-free networks to simulate the risk of traffic congestion, as shown in Fig 1.

## Risk contagion model based on information entropy

The occurrence of risk outbreaks and crisis events is an extraordinary event with a small probability. Owing to its interdisciplinarity, complexity, and uncertainty, grasping the intrinsic causes of its development, the mechanism of its evolution, and the law of contagion becomes difficult. Therefore, this study draws on the five-dimensional structure proposed to measure the magnitude of risk. The model for the five-dimensional structure of road congestion risk can be obtained as follows:

$$\mathrm{F} = (f(r_1), f(r_2), f(r_3), f(r_4), f(r_5)). \tag{1}$$

In Eq (1), $f(r_1)$ denotes the risk occurrence probability; $f(r_2)$ denotes the risk of loss; $f(r_3)$ denotes the unpredictability of risk; $f(r_4)$ denotes the uncontrollability of risk; and $f(r_5)$ denotes the transferability of risk. Because all five measures are cost-based, the higher the value of $F$, the higher the risk. Let a junction $i(i = 1, 2, \ldots, n)$ have $j(j = 1, 2, \ldots, m)$ states denoted by $s_{ij}$. When utilizing the five-dimensional metric parameters for risk measurement, a dimensionless

process is employed to ensure that the values of the five parameters lie within the [0, 1] interval, where the parameters are random variables that fall within the [0, 1] interval.

Congestion risk: Without considering the correlation between the metric parameters, the risk $R(s_{ij})$ of a junction $i$ can be defined using the geometric mean approach, as presented in Eq (2)

$$R(s_{ij}) = \sqrt[5]{f(r_1) \cdot f(r_2) \cdot f(r_3) \cdot f(r_4) \cdot f(r_5)}. \tag{2}$$

Entropy [31], used to describe the state of matter, is an important parameter for measuring the disorder of a system. The concept has been expanded and subsequent scholars have utilized this parameter to describe the movement and disorder of all matter, things, and systems. Because every system is in constant motion, entropy reflects the state of change of the structures at the micro and mesoscopic levels within a given macroscopic state; it describes the degree of disorder in the system. Generally, the higher the entropy, the more chaotic a system is, while the lower the entropy, the more orderly a system is. Therefore, the emergence and transmission of the risk of road congestion can be measured in terms of entropy to analyze the emergence and dynamics of the crisis.

Congestion risk entropy: based on the information entropy theory, the congestion risk entropy $H(s_{ij})$ of a junction $i$ can be defined as follows:

$$\begin{aligned} H(s_{ij}) &= -R(s_{ij}) \, lnR(s_{ij}) \\ &= -\sqrt[5]{f(r_1) \cdot f(r_2) \cdot f(r_3) \cdot f(r_4) \cdot f(r_5)} \\ &\quad \times ln\sqrt[5]{f(r_1) \cdot f(r_2) \cdot f(r_3) \cdot f(r_4) \cdot f(r_5)}. \end{aligned} \tag{3}$$

The correlation between the congestion risk state and entropy is established in Eq (3). Generally, a higher risk state, as measured by the risk state, corresponds to greater entropy. This indicates that the state of the road intersection was more disorderly and uncertain at that time. This method allows for a quantitative measure of the risk state of a particular intersection in the entire traffic network.

In the case of the entire traffic network, different intersections will have different positions and hence different risk capacities. Successive failure theory [32] indicates that the capacity of an intersection to handle congestion risk is related to its degree as well as the degree of all neighboring intersections. The higher the degree of the intersection and its neighboring intersections, the greater the capacity of that intersection to handle congestion risk. Moreover, different traffic networks face variations in the magnitude of congestion risk. The higher the risk faced, the greater the relative risk capacity of all intersections. Therefore, the risk capacity of $i$ is calculated as follows:

$$C_i = \frac{(d_i \sum\limits_{j \in \Gamma_i} d_j)^a}{max\left\{ (d_k \sum\limits_{i \in \Gamma_k} d_i)^a \mid k = 1, 2, \ldots, N \right\}} \bar{V}. \tag{4}$$

In Eq (4), $\alpha(0 \le \alpha \le 1)$ denotes the average degree of intersection risk capacity; the smaller its value, the more evenly distributed the intersection risk capacity and greater the variability; $d_i$ denotes the degree of the intersection $i$, $\Gamma_i$ denotes the set of neighboring intersections of the intersection $i$, $\Gamma_i = \{j|a_{ij} = 1\}$; $\bar{V}$ denotes the average congestion risk size of the entire traffic network, $\bar{V} = \sum_{i=1}^{N} H(s_{ij})p_i$ where $p_i$ is the probability of occurrence of congestion risk $H(s_{ij})$.

Most risk propagation studies rely on virus propagation dynamics models in complex networks, such as the SIS and SIR models, to investigate the phenomenon of risk propagation and its principles in various domains. However, the aforementioned models depict a chain reaction of network nodes when exposed to risk, focusing solely on transmitting the health or infection status of the node. These models ignore the fact that risk can accumulate within a node until it surpasses its tolerable threshold.

Therefore, the entropy $H(t) > C_i$ of congestion risk on junction $i$ at time $t$, where the congestion risk exceeds the bearable maximum risk resistance capacity, i.e., the risk capacity, has a certain probability of transmitting the risk to neighboring junctions and triggering their potential congestion risks. Therefore, $P(H_j|H_i)$ represents the conditional probability of the congestion risk at junction $i$ triggering the potential congestion risk at junction $j$. Therefore, the congestion risk $H_j(t + 1)$ of junction $j$ at time $t + 1$ is determined by both the congestion risk $H_j(t)$ at time $t$ and the potential congestion risk $P(H_j|H_i)$ triggered by its neighboring junctions:

$$H_j(t + 1) \quad = H_j(t) + \sum_{H_k \in \psi_j(t)} \left[ H_i \bullet OR\left\{ \phi_{lk}^{ij}(t + 1) | i \in \Gamma_j'(t), H_l \in \Phi_i(t) \right\} \right], \tag{5}$$

where $\psi_j(t)$ denotes the set of potential congestion risks at junction $j$ at time $t$,
$\psi_j(t) = \left\{ H_k \mid \eta_k^j(t) = 0 \right\}$; $\Gamma_j'(t)$ denotes the set of neighboring intersections of node $j$ at time $t$, where the risk of congestion exceeds the risk capacity, $\Gamma_i'(t) = \left\{ i | \tau_i(t) = 1, \alpha_{ij} = 1 \right\}$; $\Phi_i(t)$ denotes the set of risks that have occurred at the neighboring junctions $i (i \in \Gamma_j'(t))$ at time $t$, $\Phi_i(t) = H_l|\eta_l^j(t) = 1$; $\phi_{lk}^{ij}(t + 1)$ denotes the status of the potential risk $H_k$ of junction $j$ triggered by the congestion risk $H_l$ that has occurred at the neighboring junction $i$ at time $t + 1$. $\phi_{lk}^{ij}(t + 1) = 1$ denotes the potential risk of the triggered junction $j$ and $\phi_{lk}^{ij}(t + 1) = 0$ denotes the potential risk of untriggered junction $j$. OR is an or operator that yields a result of 1 if any of the participating set elements is 1; otherwise, it is 0. The potential congestion risk at junction $j$ is triggered when any of the neighboring junctions exceeds the risk capacity.

## Control strategies for congestion risk propagation

### Improving intersection resilience

Before the risk of congestion contagion occurs, the risk capacity of each junction is increased by external forces, such as government intervention and traffic control by the relevant authorities, to prevent congestion risk contagion. Based on this, the risk capacity of each junction $i$ is

$$C_i' = (1 + \theta)C_i, \tag{6}$$

Where $i = 1, 2, \cdots, N$, and the intersection risk resistance parameter $\theta(\theta \geq 0)$ indicates the amount of risk capacity at the intersection that can be controlled through the intervention of the government and other relevant authorities to counteract congestion risk contagion. The higher the value, the more resources the government must invest in congestion risk control at the intersection and the more risk capacity is added to each intersection. By continuously adding external forces to match $C_i'$ to the congestion risk, the intersection risk resistance parameter $\theta$ reaches a threshold $\theta^*$ as in Eq (7) (the meaning of the variables in Eq (7) is the same as in Eq (4) and will not be repeated here), indicating that the government and relevant departments invest the least amount of external resources to achieve the maximum congestion risk

resistance.

$$\theta^* = \frac{H_i(t) * max\left\{ \left( d_k \sum_{l \in \Gamma_k} d_l \right)^a \mid k = 1, 2, \ldots, N \right\} - \bar{V} * \left( d_i \sum_{j \in \Gamma_i} d_j \right)^a}{\bar{V} * \left( d_i \sum_{j \in \Gamma_i} d_j \right)^a}. \tag{7}$$

### Reducing the strength of node association

When congestion risk contagion occurs, the degree of association between intersections influences the behavior of congestion risk contagion in the entire traffic network, with varying levels of association corresponding to different levels of contagion. The degree of association between intersections $i$ and $j$ is $\omega_{ij}$. When $\omega_{ij}$ is constant, the association between junction $i$ and $j$ is linear; when $\omega_{ij}$ is a time-varying function, the coupling between junction $i$ and $j$ is nonlinear. However, in a real traffic network, the degree of risk transmission from one intersection to its neighboring intersection varies. The more critical the intersection, the stronger the degree of association with its neighboring intersection, resulting in a higher risk of congestion transmission. Consequently, the previous equation is enhanced and represented by Eqs (8), (9) and (10) as follows:

$$H_j(t + 1) = H_j(t) + \sum_{H_k \in \psi j(t)} [H_i \bullet \omega_{ij}], \tag{8}$$

$$\omega_{ij} = \frac{K_i}{Q_j}, \tag{9}$$

$$Q_j = \sum_{\mu \in \Gamma_j} K_\mu, \tag{10}$$

where $K_i$ denotes the degree of intersection $i$, $\Gamma_j$ denotes the set of neighboring intersections of intersection $j$, and $K_\mu$ denotes the degree of neighboring intersection $\mu$ of intersection $j$. Modifying the degree of association between junctions to regulate the level of congestion risk contagion decreases the risk entropy value of the junction, thereby slowing down the congestion risk transmission behavior between junctions.

## Simulation design and analysis of results

Before carrying out numerical simulations, certain parameters in the model must be set. First, a $BA$ scale-free network with $N = 1000$, $m_0 = 6$, and $m = 3$ is generated, and the five-dimensional parameter $f(r_1), f(r_2), f(r_3), f(r_4), f(r_5)$ for risk is set to a random value within $[0, 1]$. Subsequently, numerical simulations are employed to analyze the congestion risk contagion process in the network and to investigate the influence of three parameters: the risk resistance parameter $\theta$, the average degree of intersection risk capacity $\alpha$, and the degree of association $\omega_{ij}$ between intersections on the control method for congestion risk contagion.

### Analysis of congestion risk transmission processes

As each junction can bear different levels of congestion risk, when the congestion risk at a junction accumulates to a level that exceeds its risk capacity, it has a certain probability of

distributing its own risk to neighboring junctions. This results in the propagation of conges-tion risk between each individual in the network. To analyze the process of congestion risk propagation between each intersection, numerical simulations are conducted for the risk capacity $C_i$ of each intersection, the risk entropy $H(t)$ of each intersection at time $t$, and the risk entropy $H(t + 1)$ at time $t + 1$.

Fig 2 shows that the risk capacity is distributed around 0.2718. The entropy of congestion risk at time $t$ is distributed around 0.333 and at time $t + 1$ is distributed around 0.3352. It can be observed that the entropy of congestion risk at junctions changes during the propagation process. For instance, at junction 377, the congestion risk is lower than its risk capacity at time $t$. Here, no congestion risk propagation behavior occurs at this junction. However, at time $t + 1$, the congestion risk considerably surpasses its risk capacity, indicating that the neighboring junction transmits risk to this junction, resulting in an increased congestion risk at junction 377. At junction 514, the risk capacity of this junction is approximately 0.323 and the entropy of congestion risk at this junction at time $t$ is 0.347. Hence, the risk of congestion at this junc-tion is transmitted to other neighboring junctions; however, the congestion risk at time $t + 1$ is higher than that at time $t$. This indicates that while this junction propagates risk to neighboring

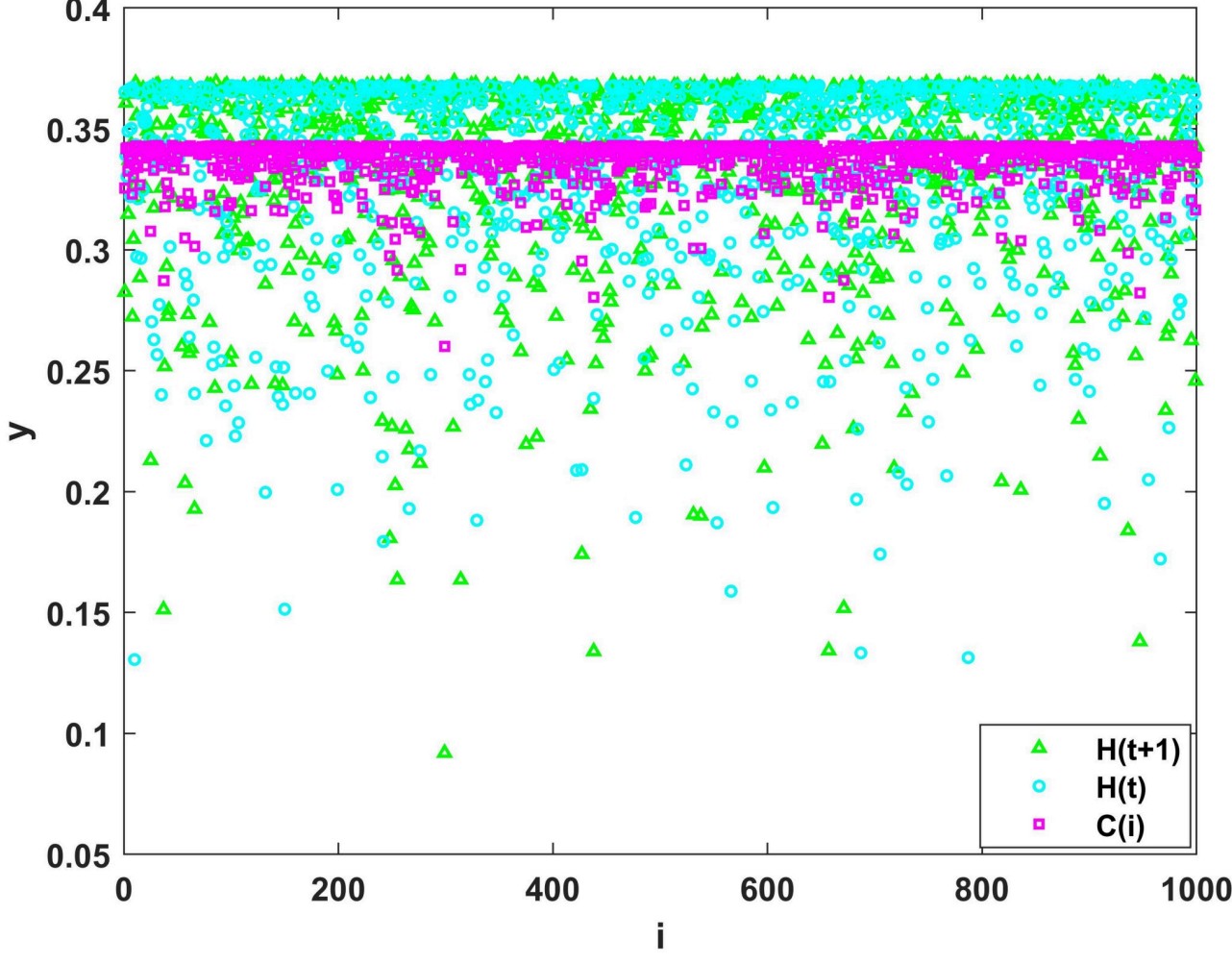

**Fig 2. Congestion risk propagation process.**

junctions, other junctions with greater risk capacity than itself will also propagate their own risk to the junction with a certain probability. As a result, the junction receives a higher entropy of congestion risk than the entropy of the risk transmitted out. At junction 527, the risk entropy at time $t + 1$ is lower than the risk entropy at time $t$, indicating that the congestion risk received by this junction is less than the risk transmitted to neighboring junctions. Therefore, this simulation result demonstrates that risk propagation at each junction in the traffic network is a complex process. When the congestion risk of a junction accumulates and exceeds its risk capacity, it will propagate its risk to neighboring junctions with a certain probability and receive congestion risk from other neighboring junctions.

The risk resistance parameter $\theta$ indicates the degree of external control exerted to prevent the propagation of congestion risk, such as government intervention or traffic control by relevant authorities. A higher value implies greater investment in risk control resources within the network, resulting in increased risk capacity at each intersection and greater risk resistance. $\alpha$ represents the average degree of intersection risk capacity. A smaller value indicates a more even distribution of risk capacity among intersections, whereas a larger value signifies greater variability. Further, to verify the influence of $\theta$ on congestion risk control, numerical simulations of the relationship between congestion risk entropy, intersection congestion risk capacity, and $\theta$ at varying $\alpha$ values of 0.15, 0.2, 0.25, and 0.3 were conducted and the results are shown in Fig 3.

Fig 3 demonstrates that, regardless of the value of $\alpha$, when the risk resistance $\theta$ is below a certain threshold, the congestion risk entropy is greater than the risk capacity, and the congestion risk is still propagated between the intersections. As $\theta$ increases, the risk capacity of each intersection increases and the congestion risk entropy decreases, indicating that the propagation of congestion risk among intersections is controlled to a certain extent. When $\theta$ exceeds a certain threshold, the risk capacity is greater than the congestion risk entropy of the intersection. Thus, the propagation of the congestion risk in the network is well controlled and the intersection gradually returns to its normal state until the congestion risk of the entire traffic network is minimized. For example, when $\alpha = 0.15$, the risk entropy is initially maximum, indicating that the congestion risk propagation has induced a traffic network failure. Here, the risk entropy decreases gradually as $\theta$ increases until $\theta \geq 4.72$. At this time, the risk entropy decreases sharply, indicating that the congestion risk propagation in the traffic network has vanished, returning the intersection gradually to its normal operation. Therefore, $\theta = 4.72$ is the threshold, called the key risk resistance threshold of the traffic network. When $\theta \geq \theta^*$, all junctions can resist the congestion risk, eliminating the occurrence of congestion risk propagation in the entire traffic network. However, when $\theta < \theta^*$, the risk capacity is limited after the increase of most junctions, congestion risk will continue accumulating at the junctions via mutual propagation, resulting in network failure. The threshold represents the ability of the network to resist the propagation of congestion risk with minimal external resources; the lower the value, the more efficient it is to control the spread of congestion risk; otherwise, it is less efficient. Therefore, the most efficient method for controlling congestion risk propagation is when the government applies an external force $\theta^*$.

Fig 3 also shows that $\theta^* = 4.72$ at $\alpha = 0.15$, $\theta^* = 4.82$ at $\alpha = 0.2$, $\theta^* = 4.88$ at $\alpha = 0.25$, and $\theta^* = 4.92$ at $\alpha = 0.3$. This indicates that as $\alpha$ increases, the key risk resistance threshold $\theta^*$ of the traffic network increases $\alpha$, indicating that the variability of the intersection risk capacity distribution increases. The only way to effectively control the occurrence of congestion risk propagation is to keep increasing the intersection risk capacity threshold, implying that congestion risk control becomes less and less efficient. Therefore, to improve the efficiency of regulating congestion risk propagation, the degree of variation in the risk capacity distribution at intersections should be considerably reduced. Thus, congestion should be carefully considered

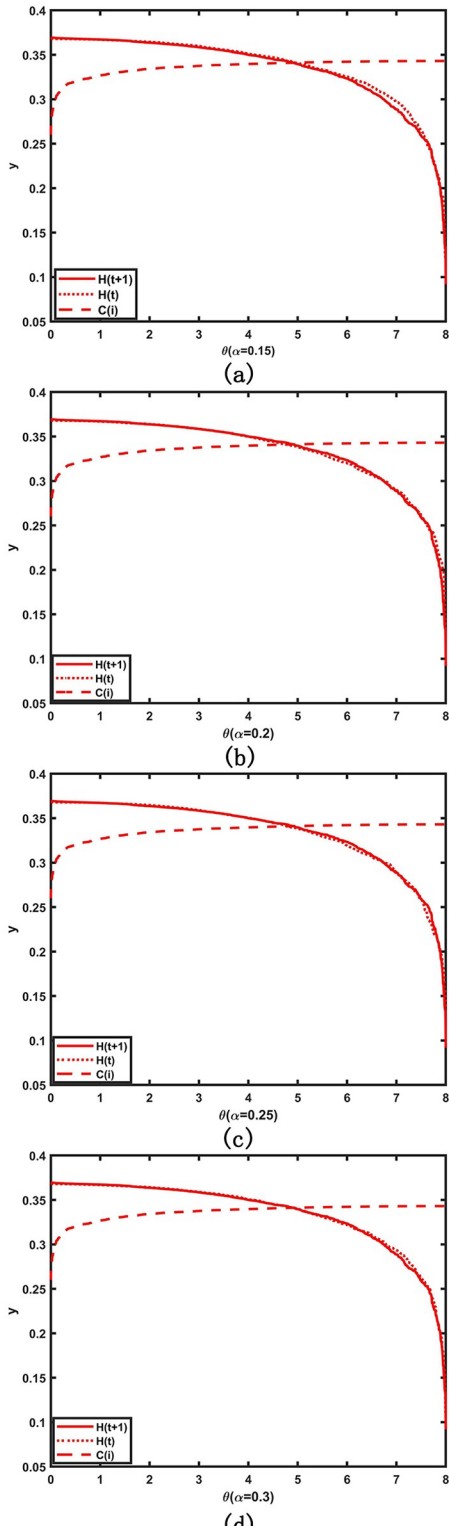

**Fig 3. Relationship between congestion risk entropy *H*, risk capacity *C* and parameter *t* for different *θ*.** (a).$\alpha =$ 0.15;(b).$\alpha = 0.2$;(c).$\alpha = 0.25$;(d).$\alpha = 0.3$.

when choosing an intersection to minimize traffic flow variations at intersections, ensuring that different intersections have approximately the same level of congestion risk resistance.

## Impact of $\omega$ on risk control

The intersection association degree $\omega$ represents the degree of mutual influence between two intersections and describes the influence of the neighboring intersections on each other. In an unweighted undirected network, $\omega$ is expressed as the ratio of the congestion risk degree to the sum of the congestion risk degrees of the neighboring intersections. Therefore, to assess the impact of $\omega$ on congestion risk control, the value of $\omega$ for each intersection is calculated based on the proposed model. The numerical simulation of $\omega$ and the congestion risk entropies at times $t$ and $t + 1$ are selected, as shown in Fig 4.

As shown in Fig 4, $H(t)$ and $H(t + 1)$ increase as $\omega$ increases, implying that the greater the value of $\omega$, the greater the risk of congestion at the intersection and greater the probability of congestion risk propagation. After $\omega$ attains a certain value, the congestion risk increases more gently at time $t$, indicating that the risk propagation peaks between the intersections at this point. As shown in Fig 3, the risk capacity is approximately 0.259 when no external force is

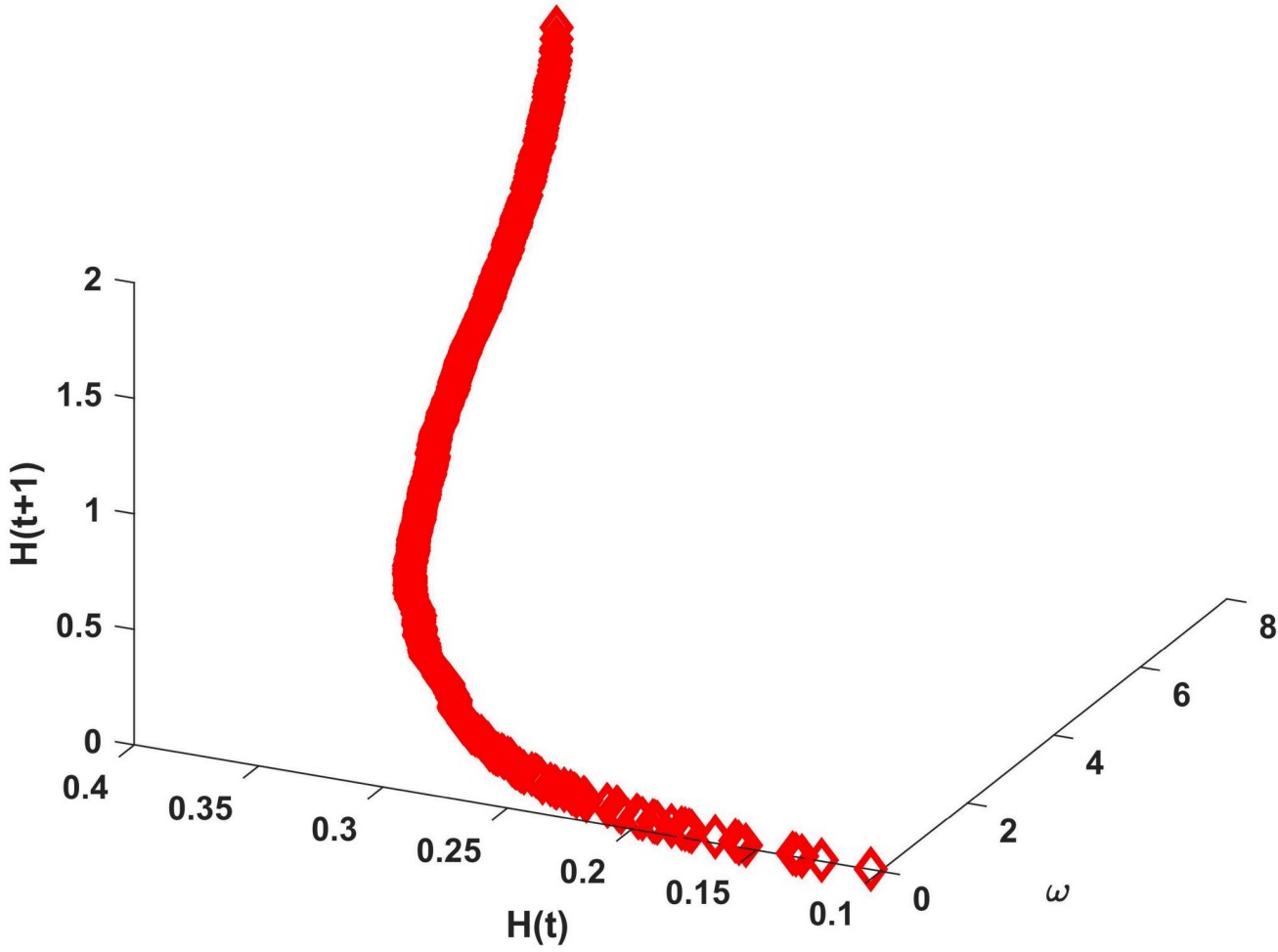

**Fig 4. Relationship between $\omega$ and $H(t)$ as well as $H(t + 1)$.**

applied. As shown in Fig 4, $\omega$ = 0.23 and the congestion risk at this time is equal to the initial risk capacity of the intersection, implying that at $\omega < 0.23$, the intersection has no congestion risk contagion. However, when $\omega$ exceeds 0.23 with the accumulation of the intersection's congestion risk, the congestion risk exceeds the risk capacity, and thus, congestion risk propagation occurs at this intersection. When $0.23 < \omega < 1.71$, the risk at time $t + 1$ is less than the risk at time $t$, indicating that the output congestion risk at the intersection is greater than the received congestion risk, resulting in decreasing risk. When $\omega = 1.71$, $H(t) = H(t + 1)$, indicating that the received congestion risk is exactly equal to the output congestion risk. When $\omega > 1.71$, the congestion risk at time $t + 1$ is greater than the congestion risk at time $t$, indicating that the congestion risk absorbed by the junction is higher than the output congestion risk. $\omega$ = 0.23 reflects the maximum degree of association the intersection can withstand to prevent congestion risk propagation. $\omega$ = 1.71 represents the minimum degree of association at which the intersection absorbs more risk than the input risk. It can be observed that the degree of association $\omega$ between intersections negatively impacts the control of congestion risk propagation. However, reducing the degree of association between intersections aids in controlling congestion risk propagation behavior in the traffic network.

## Conclusion

In this study, we propose a control method model for congestion risk propagation that takes into account the probabilities of five threats (the risk occurrence probability; the risk of loss; the unpredictability of risk; the uncontrollability of risk; the transferability of risk) in the traffic network, in terms of improving the risk resistance of intersections and reducing the degree of association among them. We employ numerical simulations to analyze the effects of the risk resistance parameter $\theta$, average degree of risk capacity of intersections $\alpha$, and degree of association $\omega$ on congestion risk control. The results reveal that risk propagation at intersections in a traffic network is relatively complex. When the congestion risk of an intersection exceeds its risk capacity, it has a certain probability of propagating its risk to neighboring intersections while also receiving congestion risk from them. Increasing the risk capacity of an intersection through external forces like government intervention or traffic control by relevant authorities does not necessarily control the propagation of congestion risk to a certain extent. However, when $\theta$ exceeds the threshold $\theta^*$, congestion risk propagation can be effectively prevented and controlled, resulting in the highest control efficiency. As the variability of risk capacity distribution at intersections increases, the occurrence of congestion risk propagation can only be effectively controlled by raising the risk capacity threshold $\theta^*$. To enhance the efficiency of controlling congestion risk propagation, minimizing the variability of the risk capacity distribution at intersections is crucial. The degree of association $\omega$ between intersections negatively impacts the efficiency of controlling congestion risk propagation. Hence, reducing the degree of association among intersections becomes essential for controlling the propagation of congestion risk in traffic networks. Based on information entropy, the risk propagation and control method proposed in this study provides a new perspective for studying urban congestion risk. Moreover, it provides new ideas for preventing and controlling the propagation of congestion risk on a larger scale. Subsequent research will concentrate on the behavioral characteristics of travelers, constructing multiple network models and conducting targeted studies on the propagation and control of different risks. This approach aims to provide a closer representation of real-world scenarios.

## Author Contributions

**Conceptualization:** Huining Yan.

**Formal analysis:** Huining Yan, Qiubai Sun, Yuxi Jiang.

**Investigation:** Qiubai Sun.

**Methodology:** Huining Yan, Hua Li, Qiubai Sun, Yuxi Jiang.

**Project administration:** Hua Li.

**Supervision:** Hua Li, Qiubai Sun, Yuxi Jiang.

**Writing – original draft:** Huining Yan.

**Writing – review & editing:** Huining Yan, Hua Li, Qiubai Sun, Yuxi Jiang.

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
