## [Decision Letter · Decision Letter 0]

6 Dec 2023

PONE-D-23-33015Propagation and Control of Congestion Risk in Scale-Free Networks Based on Information EntropyPLOS ONE

Dear Dr. Li,

Thank you for submitting your manuscript to PLOS ONE. After careful consideration, we feel that it has merit but does not fully meet PLOS ONE’s publication criteria as it currently stands. Therefore, we invite you to submit a revised version of the manuscript that addresses the points raised during the review process.

We look forward to receiving your revised manuscript.

Kind regards,

Luobing Dong

Academic Editor

PLOS ONE

Journal Requirements:

This research is funded by Natural Science Foundation of China (71771112) and Project of Liaoning Provincial Federation Social Science Circles of China (L20BGL047). 

Reviewers' comments:

Reviewer's Responses to Questions

**Comments to the Author**

1. Is the manuscript technically sound, and do the data support the conclusions?

Reviewer #1: Yes

2. Has the statistical analysis been performed appropriately and rigorously? 

Reviewer #1: I Don't Know

3. Have the authors made all data underlying the findings in their manuscript fully available?

Reviewer #1: Yes

4. Is the manuscript presented in an intelligible fashion and written in standard English?

Reviewer #1: Yes

5. Review Comments to the Author

Reviewer #1: 1. According to the authors' proposal, the congestion risk in the traffic network takes into account the probabilities of five threats (the risk occurrence probability; the risk of loss; the unpredictability of risk; the uncontrollability of risk; the transferability of risk). The model of study the propagation pattern of congestion risk in network is built on this key assumption. However, this thesis is not noted in the Conclusions and/or in the Abstract.

2. What is the meaning of the word "tr" in the sentence "Therefore, = 4.72 is the threshold, called the key risk resistance threshold tr of the traffic network." (section 4.2, penultimate paragraph)?

3. In section 3.1 the sequence of presentation is broken. It is logic to move equation (7) to the end of section 3.1.

4. Abbreviation STCS for the space-time congestion subgraph (Introduction, 2nd paragraph) is not repeated in the text.

5. Formalization of equations:

- Eq. (6) contains an unconventional three dots.

- When transferring equations that do not fit on one line, on any sign ("=" in eq. (3), "+" in eq. (5)), this sign is written at the end of the first line and the beginning of the second.

- After the equation it is customary to put a point; in cases where the next paragraph begins with "where", a comma is written.

6. PLOS authors have the option to publish the peer review history of their article (what does this mean?). If published, this will include your full peer review and any attached files.

Reviewer #1: **Yes: **Pavlo V. Anakhov

---

## [Author Response · Author response to Decision Letter 0]

3 Jan 2024

Dear Editors and Reviewers: 

Thank you for your letter and for the reviewers’ comments concerning our manuscript entitled “Propagation and Control of Congestion Risk in Scale-Free Networks Based on Information Entropy” (ID: PONE-D-23-33015). Those comments are all valuable and very helpful for revising and improving our paper, as well as the important guiding significance to our researches. We have studied comments carefully and have made correction which we hope meet with approval. In addition, we have changed the manuscript to a latex version to ensure that it meets PLOS ONE's style requirements. Revised portion are marked in yellow in the s 'Revised Manuscript with Track Changes' file.

The main corrections in the paper and the responds to the reviewer’s comments are as following:

Responds to the review’s comments:

Reviewer #1:

1.Response to comment: According to the authors' proposal, the congestion risk in the traffic network takes into account the probabilities of five threats (the risk occurrence probability; the risk of loss; the unpredictability of risk; the uncontrollability of risk; the transferability of risk). The model of study the propagation pattern of congestion risk in network is built on this key assumption. However, this thesis is not noted in the Conclusions and/or in the Abstract.

Response: Considering the Reviewer's suggestion, we have noted the congestion risk in the

traffic network takes into account the probabilities of five threats in the abstract and conclusion sections, and built a model based on this assumption. 

2. Response to comment: What is the meaning of the word "tr" in the sentence "Therefore, = 4.72 is the threshold, called the key risk resistance threshold tr of the traffic network." (section 4.2, penultimate paragraph)?

Response: We are very sorry for the negligence in writing. "tr" is redundant in the article, and we have deleted it in section 4.2. 

3. Response to comment: In section 3.1 the sequence of presentation is broken. It is logic to move equation (7) to the end of section 3.1.

Response: As suggested by the reviewer, we have moved equation (7) to the end of section 3.1.

4. Response to comment: Abbreviation STCS for the space-time congestion subgraph (Introduction, 2nd paragraph) is not repeated in the text.

Response: As Reviewer suggested that the abbreviation STCS is not repeated in the text. We have removed it from the text.

5. Response to comment: Formalization of equations:

- Eq. (6) contains an unconventional three dots.

- When transferring equations that do not fit on one line, on any sign ("=" in eq. (3),

"+" in eq. (5)), this sign is written at the end of the first line and the beginning of

the second.

- After the equation it is customary to put a point; in cases where the next

paragraph begins with "where", a comma is written.

Response: As suggested by the reviewer, we have corrected Eq. (6), changed the sign of Eq. (3) and Eq. (5), and normalized the equations throughout the text.

Special thanks to you for your good comments. We tried our best to improve the manuscript and made some changes in the manuscript. These changes will not influence the content and framework of the paper. We appreciate for Editors and Reviewers’ warm work earnestly, and hope that the correction will meet with approval. Once again, thank you very much for your comments and suggestion. 

Best regards.

Yours sincerely, 

Dr. Hua Li

---

## [Decision Letter · Decision Letter 1]

28 Feb 2024

Propagation and Control of Congestion Risk in Scale-Free Networks Based on Information Entropy

PONE-D-23-33015R1

Dear Dr. Li,

We’re pleased to inform you that your manuscript has been judged scientifically suitable for publication and will be formally accepted for publication once it meets all outstanding technical requirements.

Kind regards,

Tinggui Chen

Academic Editor

PLOS ONE

Additional Editor Comments (optional):

Reviewers' comments:

Reviewer's Responses to Questions

**Comments to the Author**

1. If the authors have adequately addressed your comments raised in a previous round of review and you feel that this manuscript is now acceptable for publication, you may indicate that here to bypass the “Comments to the Author” section, enter your conflict of interest statement in the “Confidential to Editor” section, and submit your "Accept" recommendation.

Reviewer #1: All comments have been addressed

Reviewer #2: (No Response)

2. Is the manuscript technically sound, and do the data support the conclusions?

Reviewer #1: Yes

Reviewer #2: Yes

3. Has the statistical analysis been performed appropriately and rigorously? 

Reviewer #1: I Don't Know

Reviewer #2: Yes

4. Have the authors made all data underlying the findings in their manuscript fully available?

Reviewer #1: Yes

Reviewer #2: Yes

5. Is the manuscript presented in an intelligible fashion and written in standard English?

Reviewer #1: Yes

Reviewer #2: Yes

6. Review Comments to the Author

Reviewer #1: (No Response)

Reviewer #2: General comment

I think the paper is well written, clear and fully innovative. I really appreciate it.

In particular, the authors introduce a new perspective in risk propagation, suggesting not only to consider the transmission of healthy or infection state of a node, during a shock propagation, but also the “tolerance threshold” of each node, i.e. the maximum risk resistance capacity. A conditional probability, more than a “simple” risk probability is then defined. This means that the authors take into account the fact that a dynamical system evolves and that its boundary conditions do the same. At each time step, a new system needs to be considered.

Major comments

I suggest the authors to add this “concept”, i. e. the introduction of a variable tolerance threshold for each node (intersection) to the abstract also, pushing the innovative feature of the paper.

I ask to the authors which is the time scale of variable t and t+1? I mean, on which time scale these equations are valid? If I study each minute (high frequency events) or each half-hours (low frequency events) the traffic, I suppose the relations among intersections and nodes similarities can change.

A sentence on that can be interesting to asses model robustness with respect to the temporal scale chosen for the study.

I ask the authors to read and report in the paper few “international” and famous papers of interest about both shock and disease propagation.

I suggest to start reading from:

Bardoscia, M., Caccioli, F., Perotti, J. I., Vivaldo, G., Caldarelli, G. (2016) Distress propagation in complex networks: the case of non-linear DebtRank, PLoS ONE 11(10): e0163825. DOI:10.1371/journal.pone.0163825. E-ISSN: 1932-6203.”

There the authors can find some “famous” paper about complex networks, epidemic and financial shock propagation, such as

• Pastor-Satorras R, Castellano C, Van Mieghem P, Vespignani A. Epidemic processes in complex networks. Reviews of Modern Physics. 2015;87(3):925–979.

• Van den Broeck W, Gioannini C, Gonçalves B, Quaggiotto M, Colizza V, Vespignani A. The GLEaMviz computational tool, a publicly available software to explore realistic epidemic spreading scenarios at the global scale. BMC Infectious Diseases. 2011;11:37. pmid:21288355

• Battiston S, Puliga M, Tasca P, Caldarelli G, Kaushik R, Tasca P, et al. DebtRank: too central to fail? Financial networks, the FED and systemic risk. Scientific Reports. 2012;2:541. pmid:22870377

• Musmeci N, Battiston S, Caldarelli G, Puliga M, Gabrielli A. Bootstrapping topological properties and systemic risk of complex networks using the fitness model. Journal of Statistical Physics. 2013;151(3–4):720–734.

• Barabási AL. The network takeover. Nature Physics. 2011;8(1):14–16.

I suggest to the authors for a future development of the work to consider the use of temporal networks on this model.

Minor revision: Define BA at the end of “Construction of the traffic network generation model” paragraph. I think is Barabasi-Albert, but please specify.

7. PLOS authors have the option to publish the peer review history of their article (what does this mean?). If published, this will include your full peer review and any attached files.

Reviewer #1: **Yes: **Pavlo V. Anakhov

Reviewer #2: No

---

## [Editor Report · Acceptance letter]

13 Mar 2024

PONE-D-23-33015R1 

PLOS ONE

Dear Dr. Li, 

I'm pleased to inform you that your manuscript has been deemed suitable for publication in PLOS ONE. Congratulations! Your manuscript is now being handed over to our production team.

Kind regards, 

on behalf of

Dr. Tinggui Chen 

Academic Editor

PLOS ONE